# AS-MLP: An Axial Shifted MLP Architecture for Vision

**Dongze Lian**[*], **Zehao Yu**[*]
ShanghaiTech University
{liandz,yuzh}@shanghaitech.edu.cn

**Xing Sun**
Youtu Lab, Tencent
{winfredsun}@tencent.com

**Shenghua Gao**[†]
ShanghaiTech University &
Shanghai Engineering Research Center of Intelligent Vision and Imaging &
Shanghai Engineering Research Center of Energy Efficient and Custom AI IC
{gaoshh}@shanghaitech.edu.cn

## Abstract

An Axial Shifted MLP architecture (AS-MLP) is proposed in this paper. Different from MLP-Mixer, where the global spatial feature is encoded for information flow through matrix transposition and one token-mixing MLP, we pay more attention to the local features interaction. By axially shifting channels of the feature map, AS-MLP is able to obtain the information flow from different axial directions, which captures the local dependencies. Such an operation enables us to utilize a pure MLP architecture to achieve the same local receptive field as CNN-like architecture. We can also design the receptive field size and dilation of blocks of AS-MLP, *etc*, in the same spirit of convolutional neural networks. With the proposed AS-MLP architecture, our model obtains 83.3% Top-1 accuracy with 88M parameters and 15.2 GFLOPs on the ImageNet-1K dataset. Such a simple yet effective architecture outperforms all MLP-based architectures and achieves competitive performance compared to the transformer-based architectures (*e.g.*, Swin Transformer) even with slightly lower FLOPs. In addition, AS-MLP is also the first MLP-based architecture to be applied to the downstream tasks (*e.g.*, object detection and semantic segmentation). The experimental results are also impressive. Our proposed AS-MLP obtains 51.5 mAP on the COCO validation set and 49.5 MS mIoU on the ADE20K dataset, which is competitive compared to the transformer-based architectures. Our AS-MLP establishes a strong baseline of MLP-based architecture. Code is available at https://github.com/svip-lab/AS-MLP.

## 1 Introduction

In the past decade, Convolutional Neural Networks (CNNs) (Krizhevsky et al., 2012; He et al., 2016) have received widespread attention and have become the de-facto standard for computer vision. Furthermore, with the in-depth exploration and research on self-attention, transformer-based architectures have also gradually emerged, and have surpassed CNN-based architectures in natural language processing (*e.g.*, Bert (Devlin et al., 2018)) and vision understanding (*e.g.*, ViT (Dosovitskiy et al., 2021), DeiT (Touvron et al., 2021b)) with amounts of training data. Recently, Tolstikhin et al. (2021) first propose MLP-based architecture, where almost all network parameters are learned from MLP (linear layer). It achieves amazing results, which is comparable with CNN-like models.

Such promising results drive our exploration of MLP-based architecture. In the MLP-Mixer (Tolstikhin et al., 2021), the model obtains the global receptive field through matrix transposition and token-mixing projection such that the long-range dependencies are covered. However, this rarely makes full use of the local information, which is very important in CNN-like architecture (Simonyan & Zisserman, 2015; He et al., 2016) because not all pixels need long-range dependencies, and the

---

[*]Equal contribution.
[†]Corresponding author.

local information focuses more on extracting the low-level features. In the transformer-based architectures, Swin Transformer (Liu et al., 2021b) computes the self-attention in a window ($7 \times 7$) instead of the global receptive field, which is similar to directly using a convolution layer with a large kernel size ($7 \times 7$) to cover the local receptive field. Some other papers have also already emphasized the advantages of local receptive fields, and introduced local information in the transformer, such as Localvit (Li et al., 2021), NesT (Zhang et al., 2021), *etc*. Driven by these ideas, we mainly explore the influence of locality on MLP-based architectures.

In order to introduce locality into the MLP-based architecture, one of the simplest and most intuitive ideas is to add a window to the MLP-Mixer, and then perform a token-mixing projection of the local features within the window, just as done in Swin Transformer (Liu et al., 2021b) and LINMAPPER (Fang et al., 2021). However, if we divide the window (*e.g.*, $7 \times 7$) and perform the token-mixing projection in the window, then the linear layer has the $49 \times 49$ parameters shared between windows, which greatly limits the model capacity and thus affects the learning of parameters and final results. Conversely, if the linear layer is not shared between windows, the model weights trained with fixed image size cannot be adapted to downstream tasks with various input sizes because unfixed input sizes will cause a mismatch in the number of windows.

Therefore, a more ideal way to introduce locality is to directly model the relationship between a feature point and its surrounding feature points at any position, without the need to set a fixed window (and window size) in advance. To aggregate the features of *different* spatial positions in the *same* position and model their relationships, inspired by (Wu et al., 2018; Lin et al., 2019; Wang et al., 2020; Ho et al., 2019), we propose an axial shift strategy for MLP-based architecture, where we spatially shift features in both horizontal and vertical directions. Such an approach not only aggregates features from different locations, but also makes the feature channel only need to be divided into $k$ groups instead of $k^2$ groups to obtain a receptive field of size $k \times k$ with the help of axial operation. After that, a channel-mixing MLP combines these features, enabling the model to obtain local dependencies. It also allows us to design MLP structure as the same as the convolutional kernel, for instance, to design the kernel size and dilation rate.

Based on the axial shift strategy, we design Axial Shifted MLP architecture, named AS-MLP. Our AS-MLP obtains 83.3% Top-1 accuracy with 88M parameters and 15.2 GFLOPs in the ImageNet-1K dataset without any extra training data. Such a simple yet effective method outperforms all MLP-based architectures and achieves competitive performance compared to the transformer-based architectures. It is also worth noting that the model weights in MLP-Mixer trained with fixed image size cannot be adapted to downstream tasks with various input sizes because the token-mixing MLP has a fixed dimension. On the contrary, the AS-MLP architecture can be transferred to downstream tasks (*e.g.*, object detection) due to the design of axial shift. As far as we know, it is also the first work to apply MLP-based architecture to the downstream task. With the pre-trained model in the ImageNet-1K dataset, AS-MLP obtains 51.5 mAP on the COCO validation set and 49.5 MS mIoU on the ADE20K dataset, which is competitive compared to the transformer-based architectures.

## 2 RELATED WORK

**CNN-based Architectures.** Since AlexNet (Krizhevsky et al., 2012) won the ImageNet competition in 2012, the CNN-based architectures have gradually been utilized to automatically extract image features instead of hand-crafted features. Subsequently, the VGG network (Simonyan & Zisserman, 2015) is proposed, which purely uses a series of $3 \times 3$ convolution and fully connected layers. ResNet (He et al., 2016) utilizes the residual connection to transfer features in different layers, which alleviates the gradient vanishing and obtains superior performance. Some papers make further improvements to the convolution operation in CNN-based architecture, such as dilated convolution (Yu & Koltun, 2016) and deformable convolution (Dai et al., 2017). EfficientNet (Tan & Le, 2019; 2021) introduces neural architecture search into CNN to search for a suitable network structure. These architectures build the CNN family and are used extensively in computer vision tasks.

**Transformer-based Architectures.** Transformer is first proposed in (Vaswani et al., 2017), where the attention mechanism is utilized to model the relationship between features from the different spatial positions. Subsequently, the popularity of BERT (Devlin et al., 2018) in NLP also promotes the research on transformer in the field of vision. ViT (Dosovitskiy et al., 2021) uses a transformer framework to extract visual features, where an image is divided into $16 \times 16$ patches and the convo-

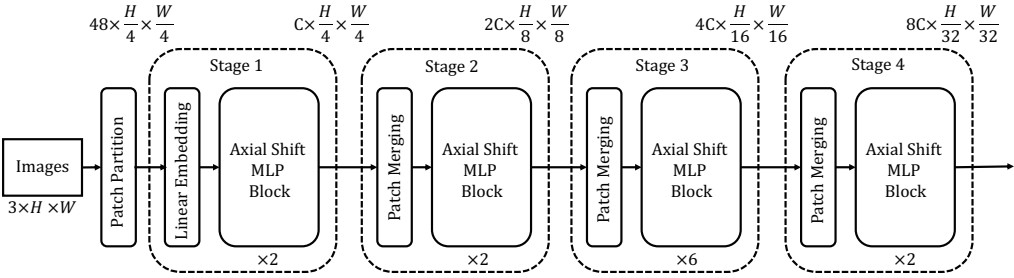

Figure 1: A tiny version of the overall Axial Shifted MLP (AS-MLP) architecture.

lution layer is completely abandoned. It shows that the transformer-based architecture can perform well in large-scale datasets (*e.g.*, JFT-300M). After that, DeiT (Touvron et al., 2021b) carefully designs training strategies and data augmentation to further improve performance on the small datasets (*e.g.*, ImageNet-1K). DeepViT (Zhou et al., 2021) and CaiT (Touvron et al., 2021c) consider the optimization problem when the network deepens, and train a deeper transformer network. CrossViT (Chen et al., 2021a) combines the local patch and global patch by using two vision transformers. CPVT (Chu et al., 2021b) uses a conditional position encoding to effectively encode the spatial positions of patches. LeViT (Graham et al., 2021) improves ViT from many aspects, including the convolution embedding, extra non-linear projection and batch normalization, *etc*. Transformer-LS (Zhu et al., 2021) proposes a long-range attention and a short-term attention to model long sequences for both language and vision tasks. Some papers also design hierarchical backbone to extract spatial features at different scales, such as PVT (Wang et al., 2021), Swin Transformer (Liu et al., 2021b), Twins (Chu et al., 2021a) and NesT (Zhang et al., 2021), which can be applied to downstream tasks.

**MLP-based Architectures.** MLP-Mixer (Tolstikhin et al., 2021) designs a very concise framework that utilizes matrix transposition and MLP to transmit information between spatial features, and obtains promising performance. The concurrent work FF (Melas-Kyriazi, 2021) also applies a similar network architecture and reaches similar conclusions. Subsequently, Res-MLP (Touvron et al., 2021a) is proposed, which also obtains impressive performance with residual MLP only trained on ImageNet-1K. gMLP (Liu et al., 2021a) and EA (Guo et al., 2021) introduce Spatial Gating Unit (SGU) and the external attention to improve the performance of the pure MLP-based architecture, respectively. Recently, Container (Gao et al., 2021) proposes a general network that unifies convolution, transformer, and MLP-Mixer. $S^2$-MLP (Yu et al., 2021) uses spatial-shift MLP for feature exchange. ViP (Hou et al., 2021). proposes a Permute-MLP layer for spatial information encoding to capture long-range dependencies. Different from these work, we focus on capturing the local dependencies with axially shifting features in the spatial dimension, which obtains better performance and can be applied to the downstream tasks. Besides, the closest concurrent work with us, CycleMLP (Chen et al., 2021b) and $S^2$-MLPv2 (Yu et al., 2021) are also proposed. $S^2$-MLPv2 improves $S^2$-MLP and CycleMLP designs Cycle Fully-Connected Layer (Cycle FC) to obtain a larger receptive field than Channel FC.

## 3 THE AS-MLP ARCHITECTURE

### 3.1 OVERALL ARCHITECTURE

Figure 1 shows our Axial Shifted MLP (AS-MLP) architecture, which refers to the style of Swin Transformer (Liu et al., 2021b). Given an RGB image $I \in \mathbb{R}^{3 \times H \times W}$, where $H$ and $W$ are the height and width of the image, respectively, AS-MLP performs the patch partition operation, which splits the original image into multiple patch tokens with the patch size of $4 \times 4$, thus the combination of all tokens has the size of $48 \times \frac{H}{4} \times \frac{W}{4}$. AS-MLP has four stages in total and there are different numbers of AS-MLP blocks in different stages. Figure 1 only shows the tiny versxion of AS-MLP, and other variants will be discussed in Sec. 3.4. All the tokens in the previous step will go through these four stages, and the final output feature will be used for image classification. In Stage 1, a linear embedding and the AS-MLP blocks are adopted for each token. The output has the dimension of $C \times \frac{H}{4} \times \frac{W}{4}$, where $C$ is the number of channels. Stage 2 first performs patch merging on the features outputted from the previous stage, which groups the neighbor $2 \times 2$ patches to obtain a feature with the size of $4C \times \frac{H}{8} \times \frac{W}{8}$ and then a linear layer is adopted to warp the feature size to $2C \times \frac{H}{8} \times \frac{W}{8}$, followed by the cascaded AS-MLP blocks. Stage 3 and Stage 4 have similar structures to Stage 2, and the hierarchical representations will be generated in these stages.

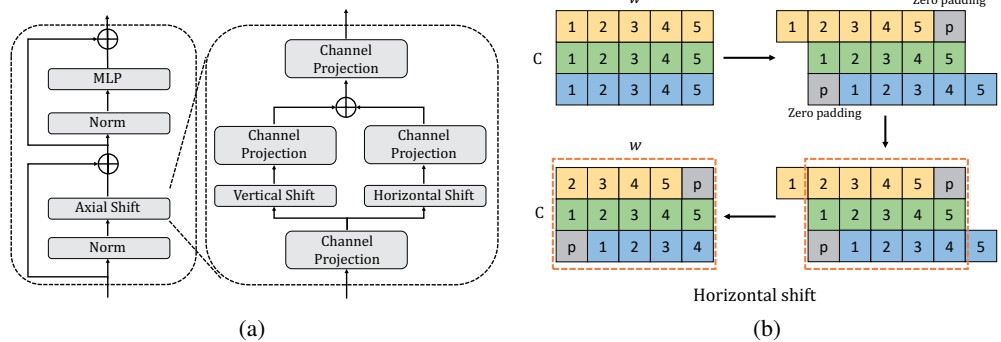

Figure 2: (a) shows the structure of the AS-MLP block; (b) shows the horizontal shift, where the arrows indicate the steps, and the number in each box is the index of the feature.

## 3.2 AS-MLP BLOCK

The core operation of AS-MLP architecture is the AS-MLP block, which is illustrated in Figure 2a. It mainly consists of the Norm layer, Axial Shift operation, MLP, and residual connection. In the Axial Shift operation, we utilize the channel projection, vertical shift, and horizontal shift to extract features, where the channel projection maps the feature with a linear layer. Vertical shift and horizontal shift are responsible for the feature translation along the spatial directions.

As shown in Figure 2b, we take the horizontal shift as an example. The input has the dimension of $C \times h \times w$. For convenience, we omit $h$ and assume $C = 3$, $w = 5$ in this figure. When the shift size is 3, the input feature is split into three parts and they are shifted by {-1, 0, 1} units along the horizontal direction, respectively. In this operation, zero padding is performed (indicated by gray blocks), and we also discuss the experimental results of using other padding methods in Sec. 4. After that, the features in the dashed box will be taken out and used for the next channel projection. The same operation is also performed in the vertical shift. In the process of both shifts, since the feature performs different shift units, the information from different spatial positions can be combined together. In the next channel projection operation, information from different spatial locations can fully flow and interact. The code of AS-MLP block is listed in Alg. 1.

**Complexity.** In the transformer-based architecture, the multi-head self-attention (MSA) is usually adopted, where the attention between tokens is computed. Swin Transformer (Liu et al., 2021b) introduces a window to partition the image and propose window multi-head self-attention (W-MSA), which only considers the computation within this window. It significantly reduces the computation complexity. However, in the AS-MLP block, without the concept of the window, we only Axially Shift (AS) the feature from the previous layer, which does not require any multiplication and addition operations. Further, the time cost of Axial Shift is very low and almost irrelevant to the shift size. Given a feature map (is usually named patches in transformer) with the dimension of $C \times h \times w$, each Axial shift operation in Figure 2a only has four channel projection operations, which has the computation complexity $4hwC^2$. If the window size in Swin Transformer (Liu et al., 2021b) is $M$, the complexities of MSA, W-MSA and AS are as follows:

$$\begin{cases} \Omega(\text{MSA}) = 4hwC^2 + 2(hw)^2C, \\ \Omega(\text{W-MSA}) = 4hwC^2 + 2M^2hwC, \\ \Omega(\text{AS}) = 4hwC^2. \end{cases} \tag{1}$$

Therefore, the AS-MLP architecture has slightly less complexity than Swin Transformer. The specific complexity calculation of each layer is shown in Appendix A.2.

## 3.3 COMPARISONS BETWEEN AS-MLP, CONVOLUTION, TRANSFORMER AND MLP-MIXER

In this section, we compare AS-MLP with the recent distinct building blocks used in computer vision, *e.g.*, the standard convolution, Swin Transformer, and MLP-Mixer. Although these modules are explored in completely different routes, from the perspective of calculation, they are all based on a given output location point, and the output depends on the weighted sum of different sampling location features (multiplication and addition operation). These sampling location features include local dependencies (*e.g.*, convolution) and long-range dependencies (*e.g.*, MLP-Mixer). Figure 3 shows the main differences of these modules in the sampling location. Given an input feature map $X \in \mathbb{R}^{H \times W \times C}$, the outputs $Y_{i,j}$ with different operations in position $(i, j)$ are as follows:

**Algorithm 1** Code of AS-MLP Block in a PyTorch-like style.

```
# norm: normalization layer
# proj: channel projection
# actn: activation layer

import torch
import torch.nn.functional as F

def shift(x, dim):
    x = F.pad(x, "constant", 0)
    x = torch.chunk(x, shift_size, 1)
    x = [ torch.roll(x_s, shift, dim) for x_s,
        shift in zip(x, range(-pad, pad+1))]
    x = torch.cat(x, 1)
    return x[:, :, pad:-pad, pad:-pad]

def as_mlp_block(x):
    shortcut = x
    x = norm(x)
    x = actn(norm(proj(x)))
    x_lr = actn(proj(shift(x, 3)))
    x_td = actn(proj(shift(x, 2)))
    x = x_lr + x_td
    x = proj(norm(x))
    return x + shortcut
```

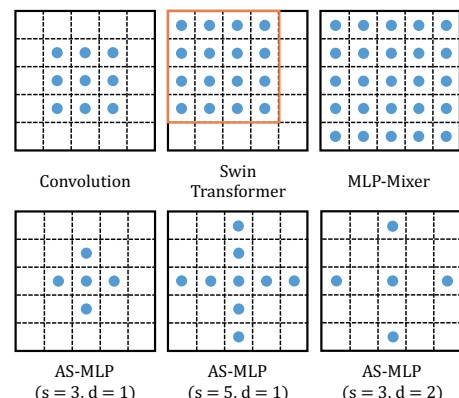

Figure 3: The different sampling locations of convolution, Swin Transformer, MLP-Mixer, and AS-MLP. *e.g.*, AS-MLP ($s = 3$, $d = 1$) shows the sampling locations when the shift size is 3 and dilation is 1.

**Convolution.** For convolution operation, a sliding kernel with the shape of $k \times k$ (receptive field region $\mathcal{R}$) is performed on $X$ to obtain the output $Y_{i,j}^{\text{conv}}$:

$$Y_{i,j}^{\text{conv}} = \sum_{(m,n)\in\mathcal{R}} X_{i+m,j+n,:} W_{m,n,:}^{\text{conv}}, \tag{2}$$

where $W^{\text{conv}} \in \mathbb{R}^{k\times k\times C}$ is the learnable weight. $h$ and $w$ are the height and width of $X$, respectively. As shown in Figure 3, the convolution operation has a local receptive field, thus it is more suitable at extracting features with the local dependencies.

**Swin Transformer.** Swin Transformer introduces a window into the transformer-based architecture to cover the local attention. The input $X$ from a window is embedded to obtain $Q, K, V$ matrix, and the output $Y_{\text{swin}}$ is the attention combination of features within the window:

$$Y_{i,j}^{\text{swin}} = \text{Softmax}(Q(X_{i,j})K(X)^T/\sqrt{d})V(X). \tag{3}$$

The introduction of locality further improves the performance of the transformer-based architecture and reduces the computational complexity.

**MLP-Mixer.** MLP-Mixer abandons the attention operation. It first transposes the input $X$, and then a token-mixing MLP is appended to obtain the output $Y_{i,j}^{\text{mixer}}$:

$$Y_{i,j}^{\text{mixer}} = (X^T W_{iW+j}^{\text{mixer}})^T, \tag{4}$$

where $W^{\text{mixer}} \in \mathbb{R}^{hw\times hw}$ is the learnable weight in token-mixing MLP. MLP-Mixer perceives the global information only with matrix transposition and MLP.

**AS-MLP.** AS-MLP axially shifts the feature map as shown in Figure 2b. Given the input $X$, shift size $s$ and dilation rate $d$, $X$ is first divided into $s$ splits in the horizontal and vertical direction. After the axial shift in Figure 2b, the output $Y_{i,j}^{\text{as}}$ is:

$$Y_{i,j}^{\text{as}} = \sum_{c=0}^{C} X_{i+\lfloor\frac{c}{\lceil C/s\rceil}\rfloor-\lfloor\frac{s}{2}\rfloor\cdot d,j,c} W_c^{\text{as-h}} + \sum_{c=0}^{C} X_{i,j+\lfloor\frac{c}{\lceil C/s\rceil}\rfloor-\lfloor\frac{s}{2}\rfloor\cdot d,c} W_c^{\text{as-v}} \tag{5}$$

where $W^{\text{as-h}}, W^{\text{as-v}} \in \mathbb{R}^C$ are the learnable weights of channel projection in the horizontal and vertical directions (here we omit activation function and bias). Unlike MLP-Mixer, we pay more attention to the local dependencies through axial shift of features and channel projection. Such an operation is closely related to Shift (Wu et al., 2018) and TSM (Lin et al., 2019). However, our method has the following characteristics: i) we use axial shift in the horizontal and vertical directions, which focuses more on the information exchange in two directions; ii) the proposed network is built upon Swin transformer and pure MLP-based architecture, where only linear layer is used and BN is replaced by LN; iii) as will be shown in Sec. 4, our network achieves superior performance, which shows the effectiveness of our method. Although such an operation can be implemented in the original shift (Wu et al., 2018), the feature channel needs to split into $k^2$ groups to achieve a receptive field of size $k \times k$. However, the axial operation makes the feature channel only need to be divided into $k$ groups instead of $k^2$ groups, which reduces the complexity.

| Network | Input Resolution | Top-1 (%) | Params | FLOPs | Throughput (image / s) |
|---|---|---|---|---|---|
| CNN-based | | | | | |
| RegNetY-8GF (Radosavovic et al., 2020) | $224 \times 224$ | 81.7 | 39M | 8.0G | 591.6 |
| RegNetY-16GF (Radosavovic et al., 2020) | $224 \times 224$ | 82.9 | 84M | 15.9G | 334.7 |
| EfficientNet-B5 (Tan & Le, 2019) | $456 \times 456$ | 83.6 | 30M | 9.9G | 169.1 |
| Transformer-based | | | | | |
| ViT-B/16 (Dosovitskiy et al., 2021) | $384 \times 384$ | 77.9 | 86M | 55.5G | 85.9 |
| DeiT-B/16 (Touvron et al., 2021b) | $224 \times 224$ | 81.8 | 86M | 17.6G | 292.3 |
| PVT-Large (Wang et al., 2021) | $224 \times 224$ | 82.3 | 61M | 9.8G | - |
| Swin-T (Liu et al., 2021b) | $224 \times 224$ | 81.3 | 29M | 4.5G | 755.2 |
| Swin-S (Liu et al., 2021b) | $224 \times 224$ | 83.0 | 50M | 8.7G | 436.9 |
| Swin-B (Liu et al., 2021b) | $224 \times 224$ | 83.3 | 88M | 15.4G | 278.1 |
| Swin-B (Liu et al., 2021b) | $384 \times 384$ | 84.2 | 88M | 47.0G | 84.7 |
| MLP-based | | | | | |
| gMLP-S (Liu et al., 2021a) | $224 \times 224$ | 79.4 | 20M | 4.5G | - |
| ViP-Small/14 (Hou et al., 2021) | $224 \times 224$ | 80.5 | 30M | - | 789.0 |
| ViP-Small/7 (Hou et al., 2021) | $224 \times 224$ | **81.5** | 25M | - | 719.0 |
| AS-MLP-T (**ours**) | $224 \times 224$ | 81.3 | 28M | 4.4G | 1047.7 |
| Mixer-B/16 (Tolstikhin et al., 2021) | $224 \times 224$ | 76.4 | 59M | 11.7G | - |
| FF (Melas-Kyriazi, 2021) | $224 \times 224$ | 74.9 | 62M | 11.4G | - |
| ResMLP-36 (Touvron et al., 2021a) | $224 \times 224$ | 79.7 | 45M | 8.9G | 478.7 |
| $S^2$-MLP-wide (Yu et al., 2021) | $224 \times 224$ | 80.0 | 68M | 13.0G | - |
| $S^2$-MLP-deep (Yu et al., 2021) | $224 \times 224$ | 80.7 | 51M | 9.7G | - |
| ViP-Medium/7 (Hou et al., 2021) | $224 \times 224$ | 82.7 | 55M | - | 418.0 |
| AS-MLP-S (**ours**) | $224 \times 224$ | **83.1** | 50M | 8.5G | 619.5 |
| gMLP-B (Liu et al., 2021a) | $224 \times 224$ | 81.6 | 73M | 15.8G | - |
| ViP-Large/7 (Hou et al., 2021) | $224 \times 224$ | 83.2 | 88M | - | 298.0 |
| AS-MLP-B (**ours**) | $224 \times 224$ | **83.3** | 88M | 15.2G | 455.2 |
| AS-MLP-B (**ours**) | $384 \times 384$ | **84.3** | 88M | 44.6G | 179.2 |

Table 1: The experimental results of different networks on ImageNet-1K. Throughput is measured with the batch size of 64 on a single V100 GPU (32GB). The more complete accuracy and throughput comparisons are listed in Appendix B.2.

## 3.4 VARIANTS OF AS-MLP ARCHITECTURE

Figure 1 only shows the tiny version of our AS-MLP architecture. Following DeiT (Touvron et al., 2021b) and Swin Transformer (Liu et al., 2021b), we also stack different number of AS-MLP blocks (the number of blocks in four stages) and expand the channel dimension ($C$ in Figure 1) to obtain variants of the AS-MLP architecture of different model sizes, which are AS-MLP-Tiny (AS-MLP-T), AS-MLP-Small (AS-MLP-S) and AS-MLP-Base (AS-MLP-B), respectively. The specific configuration is as follows:

- AS-MLP-T: $C = 96$, the number of blocks in four stages = $\{2, 2, 6, 2\}$;
- AS-MLP-S: $C = 96$, the number of blocks in four stages = $\{2, 2, 18, 2\}$;
- AS-MLP-B: $C = 128$, the number of blocks in four stages = $\{2, 2, 18, 2\}$.

The detailed configurations can be found in Appendix A.1. Table 1 in Sec. 4 shows Top-1 accuracy, model size (Params), computation complexity (FLOPs) and throughput of different variants.

## 4 EXPERIMENTS

### 4.1 IMAGE CLASSIFICATION ON THE IMAGENET-1K DATASET

**Settings.** To evaluate the effectiveness of our AS-MLP, we conduct experiments of the image classification on the ImageNet-1K benchmark, which is collected in (Deng et al., 2009). It contains 1.28M training images and 20K validation images from a total of 1000 classes. We report the experimental results with single-crop Top-1 accuracy. We use an initial learning rate of 0.001 with cosine decay and 20 epochs of linear warm-up. The AdamW (Loshchilov & Hutter, 2019) optimizer is employed to train the whole model for 300 epochs with a batch size of 1024. Following the training strategy of Swin Transformer (Liu et al., 2021b), we also use label smoothing (Szegedy et al., 2016) with a smooth ratio of 0.1 and DropPath (Huang et al., 2016) strategy.

**Results.** All image classification results are shown in Table 1. We divide all network architectures into CNN-based, Transformer-based and MLP-based architectures. The input resolution is $224 \times 224$. Our proposed AS-MLP outperforms other MLP-based architectures when keeping similar parameters and FLOPs. *e.g.*, AS-MLP-S obtains higher top-1 accuracy (83.1%) with fewer parameters than Mixer-B/16 (Tolstikhin et al., 2021) (76.4%) and ViP-Medium/7 (Hou et al., 2021) (82.7%). Furthermore, it achieves competitive performance compared with transformer-based architectures, *e.g.*, AS-MLP-B (83.3%) *vs.* Swin-B (Liu et al., 2021b) (83.3%), which shows the effectiveness of our AS-MLP architecture.

**Results of Mobile Setting.** In addition to standard experiments, we also compare the results of AS-MLP in the mobile setting, which is shown in Table 2. We build the Swin (mobile) model and AS-MLP (mobile) model with similar parameters (about 10M). The specific network details can be found in Appendix A.3. The experimental results show that our model significantly exceeds Swin Transformer (Liu et al., 2021b) in the mobile setting (76.05% *vs.* 75.11%).

| Method | Top-1 (%) | Top-5 (%) | Params |
|---|---|---|---|
| Swin (mobile) | 75.11 | 92.50 | 11.2M |
| AS-MLP (mobile) | 76.05 | 92.81 | 9.6M |

Table 2: The result comparisons of the mobile setting.

## 4.2 THE CHOICE AND IMPACT OF AS-MLP BLOCK

The core component in the AS-MLP block is the axial shift. We perform experiments to analyze the choices of different configurations of the AS-MLP block, its connection types and the impact of AS-MLP block. All ablations are conducted based on the AS-MLP-T, as shown in the setting of Sec. 3.4.

| Shift size | Padding method | d.r. | Top-1 (%) | Top-5 (%) |
|---|---|---|---|---|
| (1, 1) | N/A | 1 | 74.17 | 91.13 |
| (3, 3) | No / Circular padding | 1 | 81.04 | 95.37 |
| (3, 3) | Zero padding | 1 | 81.26 | 95.48 |
| (3, 3) | Reflect padding | 1 | 81.14 | 95.37 |
| (3, 3) | Replicate padding | 1 | 81.14 | 95.42 |
| (3, 3) | Zero padding | 2 | 80.50 | 95.12 |
| (5, 5) | Zero padding | 2 | 80.57 | 95.12 |
| (5, 5) | Zero padding | 1 | 81.34 | 95.56 |
| (7, 7) | Zero padding | 1 | 81.32 | 95.55 |
| (9, 9) | Zero padding | 1 | 81.16 | 95.45 |

| Connection type | Structure | Top-1 (%) | Top-5 (%) |
|---|---|---|---|
| Serial | $(1, 1) \rightarrow (1, 1)$ | 74.32 | 91.46 |
| | $(3, 3) \rightarrow (3, 3)$ | 81.21 | 95.42 |
| | $(5, 5) \rightarrow (5, 5)$ | 81.28 | 95.58 |
| | $(7, 7) \rightarrow (7, 7)$ | 81.17 | 95.54 |
| Parallel | $(1, 1) + (1, 1)$ | 74.17 | 91.13 |
| | $(3, 3) + (3, 3)$ | 81.26 | 95.48 |
| | $(5, 5) + (5, 5)$ | 81.34 | 95.56 |
| | $(7, 7) + (7, 7)$ | 81.32 | 95.55 |

(a) The impacts of the different configurations of the AS-MLP architecture. d.r. means dilation rate.

(b) The impacts of the different connection types. '→' means serial and '+' means parallel.

Table 3: Choices of different configurations and connection types.

**Different Configurations of AS-MLP Block.** In order to encourage the information flow from different channels in the spatial dimension, the features from the horizontal shift and the vertical shift are aggregated together in Figure 2. We evaluate the influence of different configurations of AS-MLP block, including shift size, padding method, and dilation rate, which are similar to the configuration of a convolution kernel. All experiments of different configurations are shown in Table 3a. We have three findings as follows: i) 'Zero padding' is more suitable for the design of AS-MLP block than other padding methods[1]; ii) increasing the dilation rate slightly reduces the performance of AS-MLP, which is consistent with CNN-based architecture. Dilation is usually used for semantic segmentation rather than image classification; iii) when expanding the shift size, the accuracy will increase first and then decrease. A possible reason is that the receptive field is enlarged (shift size = 5 or 7) such that AS-MLP pays attention to the global dependencies, but when shift size is 9, the network pays too much attention to the global dependencies, thus neglecting the extraction of local features, which leads to lower accuracy. Therefore, we use the configuration (shift size = 5, zero padding, dilation rate = 1) in all experiments, including object detection and semantic segmentation.

**Connection Type.** We also compare the different connection types of AS-MLP block, such as serial connection and parallel connection, and the results are shown in Table 3b. Parallel connection consistently outperforms serial connection in terms of different shift sizes, which shows the effec-

---

[1] Since we use circular shift, thus 'No padding' and 'Circular padding' are equivalent.

tiveness of the parallel connection. When the shift size is 1, the serial connection is better but it is not representative because only channel-mixing MLP is used.

**The Impact of AS-MLP Block.** We also evaluate the impact of AS-MLP block in Table 4. Here we design five baselines: i) Global-MLP; ii) Axial-MLP; iii) Window-MLP; iv) shift size (5, 1); v) shift size (1, 5). The first three baselines are designed from the perspective of how to use MLP for feature fusion at different positions, and the latter two are designed from the perspective of the axial shift in a single direction. The specific settings are listed in Appendix A.5. The results in Table 4 show that our AS-MLP block with shift size (5, 5) outperforms other baselines.

| Method | Top-1 (%) | Method | Top-1 (%) |
|---|---|---|---|
| Global-MLP | 79.81 | (5, 1) | 78.37 |
| Axial-MLP | 79.69 | (1, 5) | 78.45 |
| Window-MLP | 78.40 | (5, 5) | 81.34 |

Table 4: The impact of AS-MLP block.

| Backbone | $AP^b$ | $AP^b_{50}$ | $AP^b_{75}$ | $AP^m$ | $AP^m_{50}$ | $AP^m_{75}$ | Params | FLOPs |
|---|---|---|---|---|---|---|---|---|
| Mask R-CNN (3×) | | | | | | | | |
| ResNet50 (He et al., 2016) | 41.0 | 61.7 | 44.9 | 37.1 | 58.4 | 40.1 | 44M | 260G |
| PVT-Small (Wang et al., 2021) | 43.0 | 65.3 | 46.9 | 39.9 | 62.5 | 42.8 | 44M | 245G |
| Swin-T (Liu et al., 2021b) | 46.0 | 68.2 | 50.2 | 41.6 | 65.1 | 44.8 | 48M | 267G |
| AS-MLP-T (**ours**) | 46.0 | 67.5 | 50.7 | 41.5 | 64.6 | 44.5 | 48M | 260G |
| ResNet101 (He et al., 2016) | 42.8 | 63.2 | 47.1 | 38.5 | 60.1 | 41.3 | 63M | 336G |
| PVT-Medium (Wang et al., 2021) | 44.2 | 66.0 | 48.2 | 40.5 | 63.1 | 43.5 | 64M | 302G |
| Swin-S (Liu et al., 2021b) | 48.5 | 70.2 | 53.5 | 43.3 | 67.3 | 46.6 | 69M | 359G |
| AS-MLP-S (**ours**) | 47.8 | 68.9 | 52.5 | 42.9 | 66.4 | 46.3 | 69M | 346G |
| Cascade Mask R-CNN (3×) | | | | | | | | |
| DeiT-S (Touvron et al., 2021b) | 48.0 | 67.2 | 51.7 | 41.4 | 64.2 | 44.3 | 80M | 889G |
| ResNet50 (He et al., 2016) | 46.3 | 64.3 | 50.5 | 40.1 | 61.7 | 43.4 | 82M | 739G |
| Swin-T (Liu et al., 2021b) | 50.5 | 69.3 | 54.9 | 43.7 | 66.6 | 47.1 | 86M | 745G |
| AS-MLP-T (**ours**) | 50.1 | 68.8 | 54.3 | 43.5 | 66.3 | 46.9 | 86M | 739G |
| ResNext101-32 (Xie et al., 2017) | 48.1 | 66.5 | 52.4 | 41.6 | 63.9 | 45.2 | 101M | 819G |
| Swin-S (Liu et al., 2021b) | 51.8 | 70.4 | 56.3 | 44.7 | 67.9 | 48.5 | 107M | 838G |
| AS-MLP-S (**ours**) | 51.1 | 69.8 | 55.6 | 44.2 | 67.3 | 48.1 | 107M | 824G |
| ResNext101-64 (Xie et al., 2017) | 48.3 | 66.4 | 52.3 | 41.7 | 64.0 | 45.1 | 140M | 972G |
| Swin-B (Liu et al., 2021b) | 51.9 | 70.9 | 56.5 | 45.0 | 68.4 | 48.7 | 145M | 982G |
| AS-MLP-B (**ours**) | 51.5 | 70.0 | 56.0 | 44.7 | 67.8 | 48.4 | 145M | 961G |

Table 5: The object detection and instance segmentation results of different backbones with 3x schedule on the COCO val2017 dataset. The results with 1x schedule are listed in Appendix B.1.

## 4.3 Object Detection on COCO

The experimental setting is listed in Appendix A.4, and the results are shown in Table 5. It is worth noting that we do not compare our method with MLP-Mixer (Tolstikhin et al., 2021) because it uses a fixed spatial dimension for token-mixing MLP, which cannot be transferred to the object detection. As far as we know, we are the first work to apply MLP-based architecture to object detection. Our AS-MLP achieves comparable performance with Swin Transformer in the similar resource limitation. To be specific, Cascade Mask R-CNN + Swin-B achieves 51.9 $AP^b$ with 145M parameters and 982 GFLOPs, and Cascade Mask R-CNN + AS-MLP-B obtains 51.5 $AP^b$ with 145M parameters and 961 GFLOPs. The visualizations of object detection are shown in Appendix C.

## 4.4 Semantic Segmentation on ADE20K

The experimental setting is listed in Appendix A.4 and Table 6 shows the performance of our AS-MLP on the ADE20K dataset. Note that we are also the first to apply the MLP-based architecture to semantic segmentation. With slightly lower FLOPs, AS-MLP-T achieves better result than Swin-T (46.5 *vs.* 45.8 MS mIoU). For the large model, UperNet + Swin-B has 49.7 MS mIoU with 121M parameters and 1188 GFLOPs, and UperNet + AS-MLP-B has 49.5 MS mIoU with 121M parameters and 1166 GFLOPs, which also shows the effectiveness of our AS-MLP architecture in processing the downstream task. The visualizations of semantic segmentation are shown in Appendix C.

| Method | Backbone | val MS mIoU | Params | FLOPs |
|---|---|---|---|---|
| DANet (Fu et al., 2019a) | ResNet-101 | 45.2 | 69M | 1119G |
| DeepLabv3+ (Chen et al., 2018) | ResNet-101 | 44.1 | 63M | 1021G |
| ACNet (Fu et al., 2019b) | ResNet-101 | 45.9 | - | - |
| DNL (Yin et al., 2020) | ResNet-101 | 46.0 | 69M | 1249G |
| OCRNet (Yuan et al., 2020) | ResNet-101 | 45.3 | 56M | 923G |
| UperNet (Xiao et al., 2018) | ResNet-101 | 44.9 | 86M | 1029G |
| OCRNet (Yuan et al., 2020) | HRNet-w48 | 45.7 | 71M | 664G |
| DeepLabv3+ (Chen et al., 2018) | ResNeSt-101 | 46.9 | 66M | 1051G |
| DeepLabv3+ (Chen et al., 2018) | ResNeSt-200 | 48.4 | 88M | 1381G |
| UperNet (Xiao et al., 2018) | Swin-T (Liu et al., 2021b) | 45.8 | 60M | 945G |
| | AS-MLP-T (ours) | 46.5 | 60M | 937G |
| UperNet (Xiao et al., 2018) | Swin-S (Liu et al., 2021b) | 49.5 | 81M | 1038G |
| | AS-MLP-S (ours) | 49.2 | 81M | 1024G |
| UperNet (Xiao et al., 2018) | Swin-B (Liu et al., 2021b) | 49.7 | 121M | 1188G |
| | AS-MLP-B (ours) | 49.5 | 121M | 1166G |

Table 6: The semantic segmentation results of different backbones on the ADE20K validation set.

## 4.5 VISUALIZATION

We visualize the heatmap of learned features from Swin Transformer and AS-MLP in Figure 4, where the first column shows the image from ImageNet, and the second column shows the activation heatmap of the last layer of Swin transformer (Swin-B). The third, fourth, and fifth columns respectively indicate the response after the horizontal shift (AS-MLP (h)), the vertical shift (AS-MLP (v)) and the combination of both in the last layer of AS-MLP (AS-MLP-B). From Figure 4, one can see that i) AS-MLP can better focus on object regions compared to Swin transformer; ii) AS-MLP (h) can better focus on the vertical part of objects (as shown in the second row) because it shifts feature in the horizontal direction. It is more reasonable because the shift in the horizontal direction can cover the edge of the vertical part, which is more helpful for recognizing the object. Similarly, AS-MLP (v) can better focus on the horizontal part of objects (as shown in the fourth row).

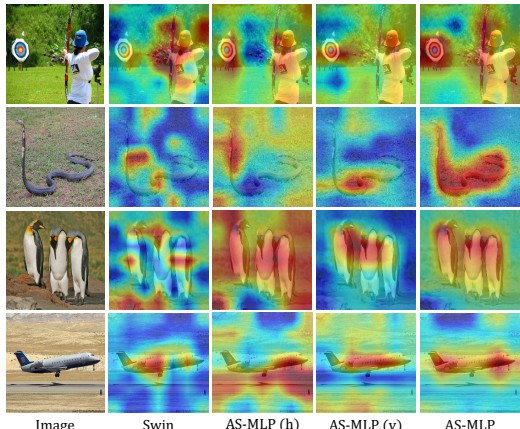

Image    Swin    AS-MLP (h)    AS-MLP (v)    AS-MLP

Figure 4: The visualization of features from Swin Transformer and our AS-MLP.

## 5 CONCLUSION AND FUTURE WORK

In this paper, we propose an axial shifted MLP architecture, named AS-MLP, for vision. Compared with MLP-Mixer, we pay more attention to the local features extraction and make full use of the channel interaction between different spatial positions through a simple feature axial shift. With the proposed AS-MLP, we further improve the performance of MLP-based architecture and the experimental results are impressive. Our model obtains 83.3% Top-1 accuracy with 88M parameters and 15.2 GFLOPs on the ImageNet-1K dataset. Such a simple yet effective method outperforms all MLP-based architectures and achieves competitive performance compared to the transformer-based architectures even with slightly lower FLOPs. We are also the first work to apply AS-MLP to the downstream tasks (*e.g.*, object detection and semantic segmentation). The results are also competitive or even better compared to transformer-based architectures, which shows the ability of MLP-based architectures in handling downstream tasks.

For future work, we will investigate the effectiveness of AS-MLP in natural language processing, and further explore the performance of AS-MLP on downstream tasks.

ACKNOWLEDGEMENT

We would like to thank Ke Li, Hao Cheng and Weixin Luo for their valuable discussions and feedback. This work was supported by National Key R&D Program of China (2018AAA0100704), NSFC #61932020, #62172279, Science and Technology Commission of Shanghai Municipality (Grant No. 20ZR1436000), and "Shuguang Program" supported by Shanghai Education Development Foundation and Shanghai Municipal Education Commission.

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

# A  THE ARCHITECTURE DETAILS

## A.1  THE DETAILED CONFIGURATIONS OF DIFFERENT ARCHITECTURES

We show the detailed configurations of different architectures in Table 7, where we assume the size of the input image is $224 \times 224$. The second column shows the output size of the image after each stage. Following Swin Transformer (Liu et al., 2021b), we use "Concat $n \times n$" to indicate a concatenation of $n \times n$ neighboring features in a patch. "shift size (5, 5)" means that the shift size in the horizontal and vertical directions is 5.

| | downsp. rate (output size) | AS-MLP-T | AS-MLP-S | AS-MLP-B |
|---|---|---|---|---|
| stage 1 | $4\times$ ($56\times56$) | concat $4\times4$, 96-d, LN  $\begin{bmatrix}\text{shift size (5, 5),} \\ \text{dim 96}\end{bmatrix} \times 2$ | concat $4\times4$, 96-d, LN  $\begin{bmatrix}\text{shift size (5, 5),} \\ \text{dim 96}\end{bmatrix} \times 2$ | concat $4\times4$, 128-d, LN  $\begin{bmatrix}\text{shift size (5, 5),} \\ \text{dim 128}\end{bmatrix} \times 2$ |
| stage 2 | $8\times$ ($28\times28$) | concat $2\times2$, 192-d, LN  $\begin{bmatrix}\text{shift size (5, 5),} \\ \text{dim 192}\end{bmatrix} \times 2$ | concat $2\times2$, 192-d, LN  $\begin{bmatrix}\text{shift size (5, 5),} \\ \text{dim 192}\end{bmatrix} \times 2$ | concat $2\times2$, 256-d, LN  $\begin{bmatrix}\text{shift size (5, 5),} \\ \text{dim 256}\end{bmatrix} \times 2$ |
| stage 3 | $16\times$ ($14\times14$) | concat $2\times2$, 384-d, LN  $\begin{bmatrix}\text{shift size (5, 5),} \\ \text{dim 384}\end{bmatrix} \times 6$ | concat $2\times2$, 384-d, LN  $\begin{bmatrix}\text{shift size (5, 5),} \\ \text{dim 384}\end{bmatrix} \times 18$ | concat $2\times2$, 512-d, LN  $\begin{bmatrix}\text{shift size (5, 5),} \\ \text{dim 512}\end{bmatrix} \times 18$ |
| stage 4 | $32\times$ ($7\times7$) | concat $2\times2$, 768-d, LN  $\begin{bmatrix}\text{shift size (5, 5),} \\ \text{dim 768}\end{bmatrix} \times 2$ | concat $2\times2$, 768-d, LN  $\begin{bmatrix}\text{shift size (5, 5),} \\ \text{dim 768}\end{bmatrix} \times 2$ | concat $2\times2$, 1024-d, LN  $\begin{bmatrix}\text{shift size (5, 5),} \\ \text{dim 1024}\end{bmatrix} \times 2$ |
| Params | | 28M | 50M | 88M |
| FLOPs | | 4.4G | 8.5G | 15.2G |

Table 7: The detailed configurations of different architectures.

## A.2  THE COMPUTATIONAL COMPLEXITY OF AS-MLP ARCHITECTURE

In this section, we show the specific computational complexity in each layer of AS-MLP architecture. The symbol definition is first given as follows. An input image: $I \in \mathbb{R}^{3 \times H \times W}$; patch size $(p, p)$; the number of blocks in four stages: $\{n_1, n_2, n_3, n_4\}$; Channel dimension $C$; MLP ratio: $r$. The specific computational complexity is shown in Table 8, where only convolution operation is computed.

| | Stage 1 | | Stage 2 | |
|---|---|---|---|---|
| | Linear embedding | AS-MLP block | Patch merging | AS-MLP block |
| Params | $3Cp^2$ | $(4+2r)C^2n_1$ | $8C^2$ | $(4+2r)4C^2n_2$ |
| FLOPs | $3Cp^2\frac{H}{p}\frac{W}{p}$ | $(4+2r)C^2\frac{H}{p}\frac{W}{p}n_1$ | $8C^2\frac{H}{2p}\frac{W}{2p}$ | $(4+2r)4C^2\frac{H}{2p}\frac{W}{2p}n_2$ |
| | Stage 3 | | Stage 4 | |
| | Patch merging | AS-MLP block | Patch merging | AS-MLP block |
| Params | $32C^2$ | $(4+2r)16C^2n_3$ | $128C^2$ | $(4+2r)64C^2n_4$ |
| FLOPs | $32C^2\frac{H}{4p}\frac{W}{4p}$ | $(4+2r)16C^2\frac{H}{4p}\frac{W}{4p}n_3$ | $128C^2\frac{H}{8p}\frac{W}{8p}$ | $(4+2r)64C^2\frac{H}{8p}\frac{W}{8p}n_4$ |

Table 8: The computational complexity of the AS-MLP Architecture.

### A.3 THE NETWORK DETAILS IN THE MOBILE SETTING

In addition to AS-MLP-T, AS-MLP-S, and AS-MLP-B, we also design AS-MLP in the mobile setting. For a fair comparison, we modify the Swin Transformer correspondingly to adopt to the mobile setting. The configurations are as follow:

- Swin (mobile): $C = 64$, the number of blocks in four stages = $\{2, 2, 2, 2\}$, the number of heads = $\{2, 4, 8, 16\}$;
- AS-MLP (mobile): $C = 64$, the number of blocks in four stages = $\{2, 2, 2, 2\}$;

### A.4 THE SETTINGS OF OBJECT DETECTION AND SEMANTIC SEGMENTATION

**Object Detection on COCO.** For the object detection and instance segmentation, we employ mmdetection (Chen et al., 2019) as the framework and COCO (Lin et al., 2014) as the evaluation dataset, which consists of 118K training data and 5K validation data. We compare the performance of our AS-MLP with other backbones on COCO. Following Swin Transformer (Liu et al., 2021b), we consider two typical object detection frameworks: Mask R-CNN (He et al., 2017) and Cascade R-CNN (Cai & Vasconcelos, 2018). The training strategies are as follows: optimizer (AdamW), learning rate (0.0001), weight decay (0.05), and batch size (2 imgs/per GPU×8 GPUs). We utilize the typical multi-scale training strategy (Carion et al., 2020; Sun et al., 2021) (the shorter side is between 480 and 800 and the longer side is at most 1333). All backbones are initialized with weights pre-trained on ImageNet-1K and all models are trained with 3x schedule (36 epochs).

**Semantic Segmentation on ADE20K.** Following Swin Transformer (Liu et al., 2021b), we conduct experiments of AS-MLP on the challenging semantic segmentation dataset, ADE20K, which contains 20,210 training images and 2,000 validation images. We utilize UperNet (Xiao et al., 2018) and AS-MLP backbone as our main experimental results. The framework is based on mmsegmentation (Contributors, 2020). The training strategies are as follows: optimizer (AdamW), learning rate ($6 \times 10^{-5}$), weight decay (0.01), and batch size (2 imgs/per GPU×8 GPUs). We utilize random horizontal flipping, random re-scaling within ratio range [0.5, 2.0] and random photometric distortion as data augmentation. The input image resolution is $512 \times 512$, the stochastic depth ratio is set as 0.3 and all models are initialized with weights pre-trained on ImageNet-1K and are trained 160K iterations.

### A.5 BASELINES

We list the specific configurations of baselines in Sec. 4.2 as follows.

- Global-MLP: following MLP-Mixer (Tolstikhin et al., 2021), we use global MLP (token-mixing MLP) along with full spatial size instead of AS-MLP block in our architecture configurations. For Global-MLP, the model weights trained with fixed image size cannot be adapted to downstream tasks with various input sizes.
- Axial-MLP: built upon Global-MLP, Axial-MLP employs two axial MLPs along with horizontal and vertical directions instead of global MLP. Similar to Global-MLP, the model weights trained with fixed image size cannot be adapted to downstream tasks with various input sizes.
- Window-MLP: as stated in Sec 1, we set fixed window ($7 \times 7$) in our architecture configurations and perform MLP operations within the window.
- Shift size (5, 1): horizontal shift is 5 and vertical shift is 1 in AS-MLP block.
- Shift size (1, 5): horizontal shift is 1 and vertical shift is 5 in AS-MLP block.

### A.6 DIFFERENCES BETWEEN AS-MLP AND TSM

We elaborate the differences between AS-MLP and TSM as follows.

- TSM performs a shift in the temporal dimension and, as stated in (Lin et al., 2019), they target the temporal dimension for efficient video understanding. However, we explore the

| Backbone | $AP^b$ | $AP^b_{50}$ | $AP^b_{75}$ | $AP^m$ | $AP^m_{50}$ | $AP^m_{75}$ | Params | FLOPs |
|---|---|---|---|---|---|---|---|---|
| Mask R-CNN (1×) | | | | | | | | |
| ResNet50 (He et al., 2016) | 38.0 | 58.6 | 41.4 | 34.4 | 55.1 | 36.7 | 44M | 260G |
| PVT-Small (Wang et al., 2021) | 40.4 | 62.9 | 43.8 | 37.8 | 60.1 | 40.3 | 44M | 245G |
| Swin-T (Liu et al., 2021b) | 43.7 | **66.6** | 47.7 | 39.8 | **63.3** | 42.7 | 48M | 267G |
| AS-MLP-T (**ours**) | **44.0** | 66.0 | **48.5** | **40.0** | 62.8 | **43.1** | 48M | 260G |
| ResNet101 (He et al., 2016) | 40.4 | 61.1 | 44.2 | 36.4 | 57.7 | 38.8 | 63M | 336G |
| PVT-Medium (Wang et al., 2021) | 42.0 | 64.4 | 45.6 | 39.0 | 61.6 | 42.1 | 64M | 302G |
| AS-MLP-S (**ours**) | **46.7** | **68.8** | **51.4** | **42.0** | **65.6** | **45.2** | 69M | 346G |
| Cascade Mask R-CNN (1×) | | | | | | | | |
| ResNet50 (He et al., 2016) | 46.3 | 64.3 | 50.5 | 40.1 | 61.7 | 43.4 | 82M | 739G |
| Swin-T (Liu et al., 2021b) | 48.1 | 67.1 | 52.2 | 41.7 | 64.4 | 45.0 | 86M | 745G |
| AS-MLP-T (**ours**) | **48.4** | **67.1** | **52.6** | **42.0** | **64.5** | **45.3** | 86M | 739G |
| AS-MLP-S (**ours**) | **50.5** | **69.4** | **54.7** | **43.7** | **66.9** | **47.3** | 107M | 824G |
| AS-MLP-B (**ours**) | **51.1** | **70.0** | **55.6** | **44.2** | **67.4** | **47.8** | 145M | 961G |

Table 9: The object detection and instance segmentation results of different backbones with 1x schedule on the COCO val2017 dataset. Mask R-CNN and Cascade Mask R-CNN frameworks are employed.

shift from a spatial perspective, for the more general tasks, such as image classification, object detection, and segmentation.

- TSM shows that shifting too many channels in a network will significantly hurt the spatial modeling ability and result in performance degradation. However, Table reftable: ablation(a) shows that as the shift size increases, the channel needs to be divided into more parts, but the performance does not decrease significantly. This suggests that the argument of TSM is not obvious in the pure MLP architecture.

- Our motivation is quite different. TSM is designed to be more effective and efficient on video. Our motivation is derived from Swin Transformer's exploration of the local receptive field of the transformer. We use shift to explore the local receptive field of MLP. Also, AS-MLP is the first MLP-based architecture for object detection and semantic segmentation with the help of such a method. Furthermore, we use axial shift to reduce the complexity of the shift split.

## B  MORE EXPERIMENTS

### B.1  MORE EXPERIMENTAL RESULTS ON COCO

Table 5 lists the object detection and instance segmentation results of different backbones with 3x schedule (36 epochs). For a complete comparison, we also conduct experiments with 1x schedule (12 epochs). The results are shown in Table 9. Our AS-MLP-T outperforms Swin-T (Liu et al., 2021b) under the Mask R-CNN (44.0 vs. 43.7 $AP^b$) and Cascade Mask R-CNN (48.4 vs. 48.1 $AP^b$) frameworks.

### B.2  THE COMPLETE CLASSIFICATION ACCURACY AND THROUGHPUT COMPARISON

Table 10 shows the complete accuracy comparison with other state-of-the-art architectures on the ImageNet-1K dataset. In addition, Table 1 shows the throughput results of the AS-MLP architecture measured with the batch size 64 on a single V100 GPU (32GB). In order to make a fair comparison with other papers, we also conduct a thorough evaluation of throughput. The results are shown in Figure 5, where we list the throughputs when the batch size is 1, 4, 8, 16, 32, 64, 128, respectively.

| Network | Input Resolution | Top-1 (%) | Params | FLOPs | Throughput (image / s) |
|---|---|---|---|---|---|
| CNN-based | | | | | |
| RegNetY-8GF (Radosavovic et al., 2020) | $224 \times 224$ | 81.7 | 39M | 8.0G | 591.6 |
| RegNetY-16GF (Radosavovic et al., 2020) | $224 \times 224$ | 82.9 | 84M | 15.9G | 334.7 |
| EfficientNet-B3 (Tan & Le, 2019) | $300 \times 300$ | 81.6 | 12M | 1.8G | 732.1 |
| EfficientNet-B5 (Tan & Le, 2019) | $456 \times 456$ | 83.6 | 30M | 9.9G | 169.1 |
| Transformer-based | | | | | |
| ViT-B/16 (Dosovitskiy et al., 2021) | $384 \times 384$ | 77.9 | 86M | 55.5G | 85.9 |
| DeiT-B/16 (Touvron et al., 2021b) | $224 \times 224$ | 81.8 | 86M | 17.6G | 292.3 |
| PVT-Large (Wang et al., 2021) | $224 \times 224$ | 82.3 | 61M | 9.8G | - |
| CPVT-B (Chu et al., 2021b) | $224 \times 224$ | 82.3 | 88M | 17.6G | 285.5 |
| TNT-B (Han et al., 2021) | $224 \times 224$ | 82.8 | 66M | 14.1G | - |
| T2T-ViT$_t$-24 (Yuan et al., 2021) | $224 \times 224$ | 82.6 | 65M | 15.0G | - |
| CaiT-S36 (Touvron et al., 2021c) | $224 \times 224$ | 83.3 | 68M | 13.9G | - |
| Swin-T (Liu et al., 2021b) | $224 \times 224$ | 81.3 | 29M | 4.5G | 755.2 |
| Swin-S (Liu et al., 2021b) | $224 \times 224$ | 83.0 | 50M | 8.7G | 436.9 |
| Swin-B (Liu et al., 2021b) | $224 \times 224$ | 83.3 | 88M | 15.4G | 278.1 |
| Nest-B (Zhang et al., 2021) | $224 \times 224$ | 83.8 | 68M | 17.9G | 235.8 |
| Container (Gao et al., 2021) | $224 \times 224$ | 82.7 | 22M | 8.1G | 347.8 |
| Swin-B (Liu et al., 2021b) | $384 \times 384$ | 84.2 | 88M | 47.0G | 84.7 |
| MLP-based | | | | | |
| gMLP-S (Liu et al., 2021a) | $224 \times 224$ | 79.4 | 20M | 4.5G | - |
| ViP-Small/14 (Hou et al., 2021) | $224 \times 224$ | 80.5 | 30M | - | 789.0 |
| ViP-Small/7 (Hou et al., 2021) | $224 \times 224$ | **81.5** | 25M | - | 719.0 |
| AS-MLP-T **(ours)** | $224 \times 224$ | 81.3 | 28M | 4.4G | 1047.7 |
| Mixer-B/16 (Tolstikhin et al., 2021) | $224 \times 224$ | 76.4 | 59M | 11.7G | - |
| FF (Melas-Kyriazi, 2021) | $224 \times 224$ | 74.9 | 62M | 11.4G | - |
| ResMLP-36 (Touvron et al., 2021a) | $224 \times 224$ | 79.7 | 45M | 8.9G | 478.7 |
| $S^2$-MLP-wide (Yu et al., 2021) | $224 \times 224$ | 80.0 | 68M | 13.0G | - |
| $S^2$-MLP-deep (Yu et al., 2021) | $224 \times 224$ | 80.7 | 51M | 9.7G | - |
| ViP-Medium/7 (Hou et al., 2021) | $224 \times 224$ | 82.7 | 55M | - | 418.0 |
| AS-MLP-S **(ours)** | $224 \times 224$ | **83.1** | 50M | 8.5G | 619.5 |
| gMLP-B (Liu et al., 2021a) | $224 \times 224$ | 81.6 | 73M | 15.8G | - |
| ViP-Large/7 (Hou et al., 2021) | $224 \times 224$ | 83.2 | 88M | - | 298.0 |
| AS-MLP-B **(ours)** | $224 \times 224$ | **83.3** | 88M | 15.2G | 455.2 |
| AS-MLP-B **(ours)** | $384 \times 384$ | **84.3** | 88M | 44.6G | 179.2 |

Table 10: The complete experimental results of different networks on ImageNet-1K. Throughput is measured with the batch size of 64 on a single V100 GPU (32GB).

## B.3 Evaluation Accuracy

In Figure 6, we visualize the evaluation accuracy of AS-MLP and Swin transformer on the ImageNet, COCO and ADE20K datasets during training. For image classification on ImageNet, AS-MLP-T keeps pace with Swin-T in each epoch and they finally converge to the similar accuracy (81.3 vs. 81.3). For object detection and semantic segmentation on COCO and ADE20K, we can see that AS-MLP-T achieves better performance than Swin-T in the early stage, and keeps winning during the training process.

## B.4 Comparisons to ShiftResNet

In this section, we compare the performance with ShiftResNet (Wu et al., 2018) in Table 11. The results of ShiftResNet50 with different configurations are from Table 3 of ShiftResNet paper (Wu et al., 2018). Our AS-MLP (mobile) achieves better accuracy with fewer parameters.

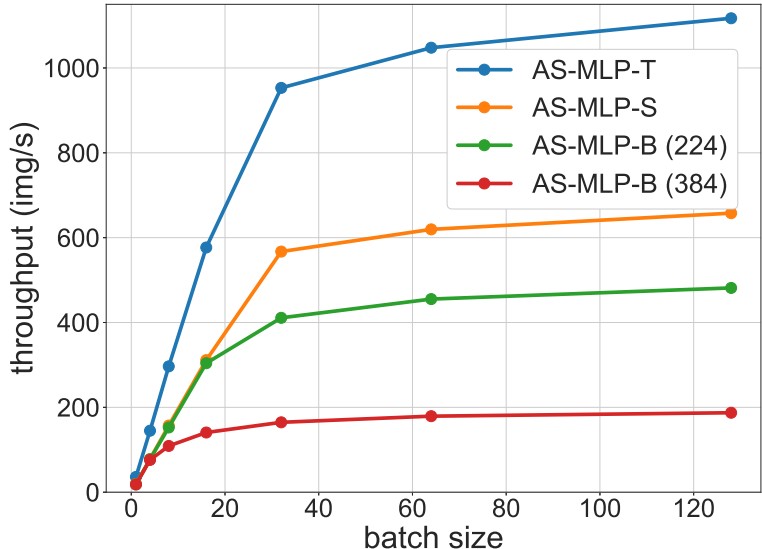

Figure 5: The throughput curve when the batch size is 1, 4, 8, 16, 32, 64, 128, respectively.

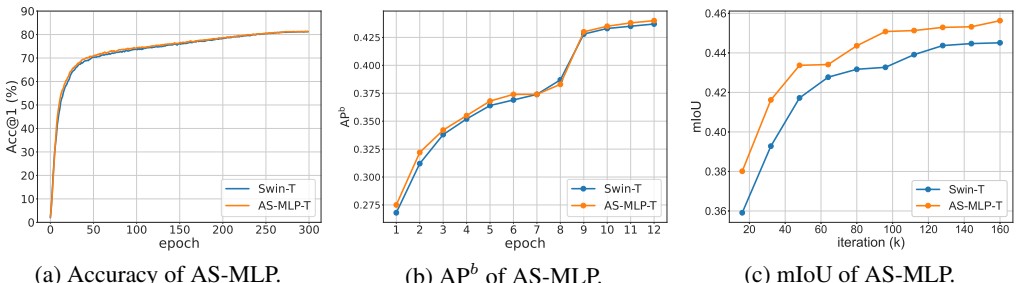

(a) Accuracy of AS-MLP.  (b) $AP^b$ of AS-MLP.  (c) mIoU of AS-MLP.

Figure 6: The evaluation accuracy of AS-MLP and Swin transformer on the ImageNet, COCO and ADE20K datasets during training.

## C  THE VISUALIZATION OF RESULTS ON COCO AND ADE20K

We visualize the object detection and instance segmentation results on the COCO dataset in Figure 7, where the Cascade Mask R-CNN model with the AS-MLP-T backbone is used. We also visualize the semantic segmentation results on the ADE20K dataset in Figure 8, where we utilize the UperNet model with the AS-MLP-T backbone. The object can be detected and segmented correctly.

| Method | Top-1 (%) | Top-5 (%) | Params |
|---|---|---|---|
| ShiftResNet50-0 | 70.6 | 89.9 | 6M |
| ShiftResNet50-1 | 73.7 | 91.8 | 11M |
| ShiftResNet50-2 | 75.6 | 92.8 | 22M |
| AS-MLP (mobile) | 76.05 | 92.81 | 9.6M |
| AS-MLP-T | 81.34 | 95.56 | 28M |

Table 11: The comparisons with ShiftResNet.

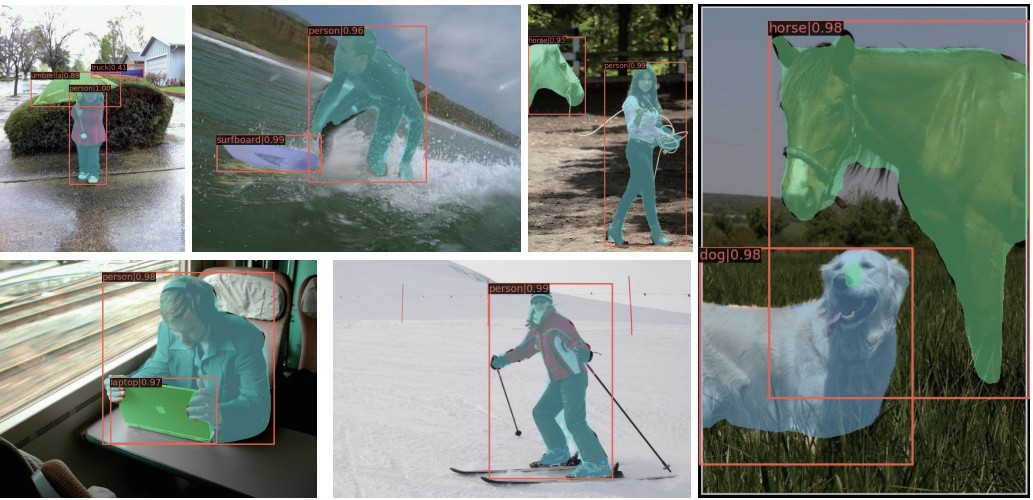

Figure 7: The object detection and instance segmentation results on the COCO dataset.

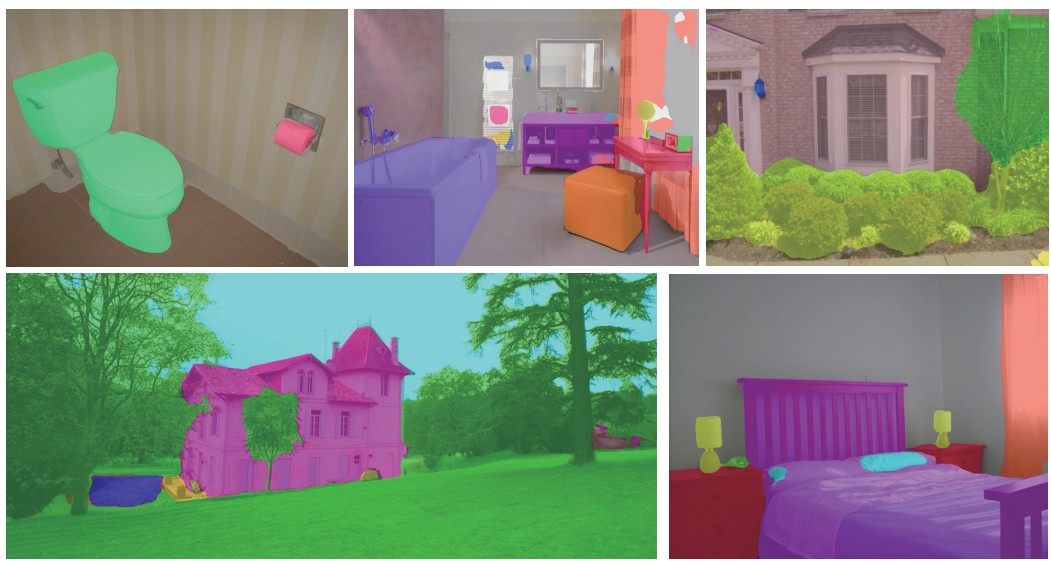

Figure 8: The semantic segmentation results on the ADE20K dataset.

