# OpenReview forum: "AS-MLP: An Axial Shifted MLP Architecture for Vision"
_ICLR.cc/2022/Conference — ICLR 2022 Poster_

### Official Review · Reviewer_Wh4q · 2021-10-30

**Correctness:** 3
**Technical Novelty And Significance:** 3
**Empirical Novelty And Significance:** 4
**Recommendation:** 5
**Confidence:** 4

**Main Review:**

Strengths
1. The paper is well written and presented clearly.
2. The proposed axial shift module is simple and elegant.
3. Experiments are done extensively beyond ImageNet.

Weaknesses
1. Limited technical novelty given the fact that the shift operation (CVPR 2018) was already presented as an alternative (though similar) operation to convolution. The shift-ResNet is already spatial convolution free except the first stem layer. Now it is not surprising that using a convolution alternative in MLP-mixer can boost its performance to convolution level (i.e. better than previous MLP-mixers).
2. The modification of using “axial” on top of “shift” lacks justification or ablation. What if the original shift operation is applied to the nearby 8 points in the 3x3 window? The channels can be kept the same by replacing “horizontal shift” with 4 points and “vertical shift” with another 4 points. This should be very similar to the “5x5” configure used in the paper.
3. Another intuitive baseline is simply to use depth wise convolution with a depth multiplier of 2, instead of the 2 parallel axial shift module. Would this simple convolutional baseline affect performance/params/FLOPs/throughput by a large margin? Possibly not. If this simple convolutional baseline works, then what is the benefit of using AS-MLP-mixers?
4. Efficientnetv2 (ICML 2021), a representative convolution-based method is not cited/discussed/compared. For example, Table 11 of Efficientnetv2 presented V2-S with 8.8G FLOPs, 901 images/s, 83.6% top-1. This is better than AS-MLP-B, 15.2G FLOPs, 455.2 images/s. Although the throughput comparison is not 100% fair (Efficientnetn2 uses a larger batch size, FP16 inference, and SE module), I think AS-MLP-B can hardly perform better than Efficientnetv2 in a fair throughput comparison.

**Summary Of The Paper:**

This paper proposes to use the shift operation (Wu et al. CVPR 2018) in an axial manner for MLP-mixer architectures. The proposed method performs much better than previous MLP-based methods on ImageNet-1K and on par with Swin-transformer.

**Summary Of The Review:**

I think this paper is around the acceptance threshold, mainly due to the limited technical novelty besides applying Shift, an alternative to convolution, to MLP-mixers, and achieving on par performance as previous transformer/convolution methods. I will raise my score if the weaknesses are addressed.

---

> ### Author Response · Authors · 2021-11-16
> **Response to Reviewer Wh4q (1/2)**
>
> Thank you for your careful and insightful comments. In the response, we have tried our best to address the questions. And the paper has also been modified accordingly.
>
> Q1: ‘Limited technical novelty given the fact that the shift operation (CVPR 2018) was already presented as an alternative (though similar) operation to convolution. The shift-ResNet is already spatial convolution free except the first stem layer. Now it is not surprising that using a convolution alternative in MLP-mixer can boost its performance to convolution level (i.e. better than previous MLP-mixers).’
>
> A1: Thank you for your suggestion. We have listed the differences between AS-MLP and Shift in the Sec. 3.3 of the original version. We also add comparisons to ShiftResNet and Table 11 in Appendix B.4 of the revised version. As shown in the Shift paper, their method achieves comparable accuracy to convolution but with fewer parameters. MLP-Mixer uses the global receptive field. Perhaps it is not surprising that using shift operation on MLP-mixer can boost its performance, but our results can also achieve comparable performance with existing transformers, which is not obvious. It is attributed to the architecture design and AS-MLP block.
>
> Q2: ‘The modification of using “axial” on top of “shift” lacks justification or ablation. What if the original shift operation is applied to the nearby 8 points in the 3x3 window? The channels can be kept the same by replacing “horizontal shift” with 4 points and “vertical shift” with another 4 points. This should be very similar to the “5x5” configure used in the paper.’
>
> A2: Thanks for your suggestion. We have not elaborated clearly in the original version. The motivation of using ‘axial’ is to reduce complexity. In the original shift operation, if we apply shift to a 3x3 window, we need to shift to the nearby 8 points and the channel needs to be divided into 9 groups, as the Shift paper states. To achieve a receptive field of 5x5, the channels would need to be divided into 25 groups. Similarly for 7x7 and 9x9, more groupings are needed, typically O(k^2) for a receptive field of kxk, which is complex. If the first layer has 96 or 64 channels, and we need the receptive field of 7x7 or 9x9 (e.g., in semantic segmentation), the number of channels is not enough or the model achieves a sub-optimal result. However, such a large number of groups might not be necessary, and when using axial shift, only k groups are needed and the same field size is achieved. As you say, if the original shift is applied to 8 points in a 3x3 window, it is equivalent to our configuration (3, 3), not to (5, 5). As can be seen in Figure 3, our (5, 5) has a larger receptive field. These contents are elaborated in Sec. 1 and Sec. 3.3 of the revised version.
>
>
> Q3: ‘Another intuitive baseline is simply to use depth-wise convolution with a depth multiplier of 2, instead of the 2 parallel axial shift module. Would this simple convolutional baseline affect performance/params/FLOPs/throughput by a large margin? Possibly not. If this simple convolutional baseline works, then what is the benefit of using AS-MLP-mixers?’
>
> A3: First of all, our work focus on exploring the performance of MLP-like architecture. Depthwise convolution still falls under the category of convolution, as does the MLP-Mixer, although its performance is not as good as that of transformer-based architecture, its experiments are impressive. To push it forward, we bring the performance of MLP-like architecture up to the same level as the transformer. Regarding your proposed baseline, we conduct an experiment built upon AS-MLP-T. We use the block (1x1conv -> 3x3 depthwise conv -> 1x1 conv) instead of AS-MLP block to hold similar parameters. It achieves 80.7% Top-1 accuracy, which is still lower than AS-MLP-T (81.3%).

---

> ### Author Response · Authors · 2021-11-16
> **Response to Reviewer Wh4q (2/2)**
>
> Q4: ‘Efficientnetv2 (ICML 2021), a representative convolution-based method is not cited/discussed/compared. For example, Table 11 of Efficientnetv2 presented V2-S with 8.8G FLOPs, 901 images/s, 83.6% top-1. This is better than AS-MLP-B, 15.2G FLOPs, 455.2 images/s. Although the throughput comparison is not 100% fair (Efficientnetn2 uses a larger batch size, FP16 inference, and SE module), I think AS-MLP-B can hardly perform better than Efficientnetv2 in a fair throughput comparison.’
>
> A4: Sorry for the missing reference. We have cited Efficientnetv2 in the revised version. As shown in Table 7 and Table 11 of the Efficientnetv2 paper, AS-MLP-B achieves slightly lower accuracy than Efficientnetv2 (83.3% vs. 83.6%). However, we would like to argue that it is not fair to compare performance with Efficientnetv2. Efficientnetv2 uses NAS to search for a better architecture. As shown in Table 5 of Autoformer (NeurIPS 2021), using NAS can further improve the performance of transformer-based architecture. Otherwise, Efficientnetv2 uses some other strategies such as progressive learning. These strategies can also be added to the AS-MLP. In such a short rebuttal time, it is hard to conduct experiments such as NAS or progressive learning. On the contrary, our comparisons with Swin transformer and other MLP-based architectures are fair. The complete comparisons are shown in Table 1. AS-MLP achieves competitive performance compared with transformer-based architectures with slightly fewer FLOPs and higher throughput, and also outperforms other MLP-based architectures.
>
> If we have misunderstood your questions or you have any other concerns, please feel free to reply. Your constructive comments are very important for the enhancement of our manuscript.

---

> ### Author Response · Authors · 2021-11-25
> **Response to Reviewer Wh4q**
>
> Dear Reviewer Wh4q,
>
> Do our previous responses and revised manuscript address your concerns? Please feel free to let us know if you have other questions.

---

> ### Comment · Reviewer_Wh4q · 2021-11-30
> **Response to the authors**
>
> Thank you for the feedback and for the updated manuscript. However, the response does not sufficiently address my concerns, so I keep my original rating of 5. The main concern is still limited justified novelty.
>
> The novel design of axial-shift lacks support because non-axial 3x3 will use the same number of groups (8) as the axial 5x5 design, but the performance looks similar for axial 3x3 (Table 3), axial 5x5 (Table 3), and presumably non-axial 3x3. Without this axial-shift design, the proposed operation is again similar to Shift (CVPR 2018, which uses a weak recipe, making table 11 unfair), an alternative to convolution. Moreover, convolutional Efficientnetv2 (admittedly with a stronger recipe and NAS) with half the size performs better than the proposed method and swin as well. This makes the contribution of "making MLP competitive to swin" less surprising, especially when the proposed method uses a convolution alternative.

---

> > ### Author Response · Authors · 2021-11-30
> > **Response to Reviewer Wh4q**
> >
> > Thanks for your response. Our new response is as follows.
> >
> > Q1: non-axial 3x3 will use the same number of groups (8) as the axial 5x5 design, but the performance looks similar for axial 3x3 (Table 3), axial 5x5 (Table 3), and presumably non-axial 3x3.
> >
> > A1: It is worth noting that although non-axial 3x3 will use the same number of groups (8) as the axial 5x5 design, shift (3, 3) is not equivalent to shift (5, 5), because shift (5, 5) has a larger receptive field. For image classification, the model might not require a large receptive field, thus shift (3, 3) has a similar accuracy with shift (5, 5). For downstream tasks such as semantic segmentation, as we state in the first reply, segmentation requires a large receptive field to improve performance [1]. We also conduct experiments in AS-MLP-T. AS-MLP-T achieves 44.4 miou and 45.5 ms miou with shift (3, 3), and 45.6 miou and 46.5 ms miou with shift (5, 5), respectively. The improvement shows the effectiveness of a large receptive field for segmentation.
> >
> > [1] Chao Peng, Xiangyu Zhang, Gang Yu, Guiming Luo and Jian Sun. Large Kernel Matters ——Improve Semantic Segmentation by Global Convolutional Network, CVPR2017.

---

### Official Review · Reviewer_uLzi · 2021-10-31

**Correctness:** 3
**Technical Novelty And Significance:** 3
**Empirical Novelty And Significance:** 3
**Recommendation:** 6
**Confidence:** 4

**Main Review:**

Strengths:
* The approach is an interesting combination of existing ideas.
* The presented experimental results are competitive and cover three tasks.

Weaknesses
* The novelty of the approach seems somewhat limited. Taking Swin, MLP-Mixer (and colleagues), and Shift/TSM/ViP/S2-MLP, the added delta seems not very large. The main differences are highlighted at the end of Section 2 ("we focus on capturing the local dependencies with axially shifting features in the spatial dimension, which obtains better performance and can be applied to the downstream tasks.") and at the end of Section 3.3. (as (i) through (iii)). None of these differences is very convincing in my opinion, specifically the claimed "better performance" is not obvious after looking at all the result tables in detail.
* The relation to both plain MLP-Mixer and to convolution is not clearly shown experimentally. It seems like both convolution and MLP-Mixer could appear as a limiting case of (some form of) AS-MLP. It would be great to understand this better both in principle and through experimental comparisons.
* The main idea of the "axial shift" is not introduced with sufficient clarity in my opinion. Even after carefully paying attention to Sections 3.2 and 3.3, Figures 2 and 3, Algorithm 1, (and briefly looking at the code), I still had some doubts as to whether I have understood all details correctly. (Of course that could be my fault, I'm curious what the other reviewers think.) In my opinion, the notation used in Sec 3.3. is also slightly incomplete.

Minor points:

In the introduction, the "axial shift strategy" is mentioned, but the explanation remains very vague and the reader has to wait until Section 3.2 to get a more detailed description. I think it would be better to introduce this main idea earlier and with more clarity.

"enables the model to obtain more local dependencies, thus improve the performance" - The "thus" here is not a priori given in my opinion. At least it would require more elaboration.

"the first work to apply MLP-based architecture to the downstream task" - There seems to be a strong relation to the recent CycleMLP, which may be concurrent work, but should nevertheless be cited and contrasted, I think.
Chen  et al.: CycleMLP: A MLP-like Architecture for Dense Prediction, arXiv:2107.10224, 2021

Footnote 2: This is a very strong statement ("cannot"). It is probably clear that this does not work "out of the box", but e.g. the MLP-Mixer paper describes how an increase in image resolution can be handled (Appendix C). At least this statement could be discussed and explained in more detail.

4.1 Settings: among the myriad of regularization strategies that exist, why are specifically label smoothing and DropPath chosen? How many other regularization strategies were tried? Or maybe is this based on Swin, in which case it should be noted here.

4.2 This Section seems more like a parameter choice discussion than an ablation study to me.

4.5 The statements derived from the four examples in Figure 4 ("From Figure 4, one can see that") seem like overclaiming, at least in parts: "These phenomena further show the superiority of AS-MLP."

In the context of "axial attention" (even though the AS-MLP does not use attention) it seems there are at least two papers that are quite well-known and it might be useful to add them to the related work:
Wang et al.: Axial-DeepLab [...], ECCV, 2020
Ho et al.: Axial Attention in multidimensional transformers, arXiv:1912.12180, 2019

Minor details (not influencing the recommendation):
* Grammar: There are several errors in English grammar, the most prominent category seems the use of articles (definite "the" vs. indefinite "a", vs. no article).
* "firstly" - in most cases where this word is used in the paper I think it should just be "first"
* Footnote 1: I did not understand this footnote.
* Algorithm 1: x_c -> x_s
* "uses a conditional position encoding to effectively encodes" -> encode
* Fig.1 seems to be heavily based on Swin Transformer's Fig. 3a - maybe cite that paper here? Similarly for the specific configurations in Sec 3.4.
* "The introduction of locality further improves the performance of the transformer-based architecture
and reduce the computational complexity." -> reduces
* References:
** Some arXiv citations have appeared in conferences, e.g. the Transformer paper was published in NeurIPS or ResNet in CVPR, I think. It would be nice to cite the conference version for such papers (I did not check all).
** Some words (mostly abbreviation) in reference titles should not be lowercase, e.g. "r-cnn", "mlp", "Resmlp"
** Consistency: MMDetection has >10 authors listed, but other papers stop after the first N authors with "et al."
* Figure 7 and 8, while looking ok when viewed on screen, turned out very garbled in the color space when printed on paper, it would be useful to check this.

--------------------------

Update after reading other reviewers' comments, the authors' comments, and considering the updated paper:

All three reviewers seem to have a similar view of the submitted work and agree in their rating. I think the updated paper and the authors' comments have addressed some of the concerns that the reviewers have raised. I think that the updated paper has improved in quality. On the other hand I think that some of the weaknesses still remain. Overall, I believe that the paper is close to the acceptance threshold. Seeing the improvements to the paper I am willing to raise my score to "marginally above the acceptance threshold".

**Summary Of The Paper:**

The paper proposes a new architecture for computer vision that is inspired by (a) the Swin Transformer, (b) MLP-Mixer (and colleagues) and (c) CNN-like local context via shifts (like Shift, TSM, ViP, S2-MLP). The architecture is based on the Swin Transformer, removes the windowed-attention, and then adds "local shifts" of channels to introduce local context via MLP. The architecture is applied to ImageNet-1k classification, COCO detection, and ADE20k segmentation with good results relative to model size and performance.


**Summary Of The Review:**

An interesting combination of existing ideas evaluated on three computer vision tasks. Overall novelty seems limited, the paper does not explain its approach very well, and the gained understanding is limited. Empirical results are interesting but not super-convincing.

---

> ### Author Response · Authors · 2021-11-16
> **Response to Reviewer uLzi (1/2)**
>
> Thank you for your valuable comments. We have modified our main text in the revised version according to your questions and suggestions. Please see the response for details.
>
> Q1: ‘The novelty of the approach seems somewhat limited. Taking Swin, MLP-Mixer (and colleagues), and Shift/TSM/ViP/S2-MLP, the added delta seems not very large. Specifically the claimed "better performance" is not obvious after looking at all the result tables in detail.’
>
> A1: Regarding the differences with Shift, as you commented, we have already described it in the original version and also cited TSM paper. The specific differences between AS-MLP and TSM refer to Response to HTjG, please. Regarding Vip and S2-MLP, according to the ICLR review rules (https://iclr.cc/Conferences/2022/ReviewerGuide), they are considered concurrent work. To contain complete related papers, we also discuss their specific methods and experimental results in the original version. Regarding the better performance, we would like to emphasize that compared with Shift, whose performance is listed in the appendix of the revised version for your suggestion, AS-MLP achieves better performance. In addition, if we regard Vip and S2-MLP as the concurrent work, for example, in Table 1, AS-MLP also achieves the best performance, and it is comparable to transformer-based architecture, such as Swin transformer. Our AS-MLP can also be easily transferred to downstream tasks and it is the first architecture for object detection and segmentation. This is also a difference from the previous work.
>
> Q2: ‘It seems like both convolution and MLP-Mixer could appear as a limiting case of (some form of) AS-MLP. It would be great to understand this better both in principle and through experimental comparisons.’
>
> A2: The convolution and MLP-Mixer are not the limiting cases of AS-MLP. The main difference is different sampling locations. We have clarified this in Sec. 3.3 and Figure 3 in the revised version.
>
> Q3: ‘The main idea of the "axial shift" is not introduced with sufficient clarity in my opinion. In my opinion, the notation used in Sec 3.3. is also slightly incomplete.’
>
> A3: Thank you for your suggestion. It might cause a misunderstanding to you due to our unclear statement. In the revised version, we elaborate it in Sec. 1. Similar to Swin, it explores the influence of local receptive fields based on Vit. We also do such an exploration based on MLP-Mixer. The above exploration on the importance of the local receptive field shows impressive results. The specific motivation is as follows: to introduce locality into the MLP-based architecture, one of the simplest and most intuitive ideas is to add a window to the MLP-Mixer, and then perform a token-mixing projection of the local information on the features within the window. However, for the MLP-based architecture, if we divide the window (e.g., 7x7) and perform the token-mixing projection in the window, then the linear layer has the 49x49 parameters shared between windows, which greatly limits the model capacity and thus affects the learning of parameters and the final results. Therefore, a more ideal way to introduce locality is to directly model the relationship between a feature point and its surrounding feature points at any position, without the need to set a fixed window (and window size) in advance. To aggregate the features of different spatial positions in the same position and model their relationships, we propose an axial shift strategy for MLP-based architecture, where we spatially shift features in both horizontal and vertical directions. Such an approach not only aggregates features from different locations, but also makes the feature channel only need to be divided into k groups instead of k^2 groups to obtain a receptive field of size kxk (Please also refer to A2 of response to Wh4q) with the help of axial operation. After that, a channel-mixing MLP combines these features, enabling the model to obtain local dependencies.
> Regarding Sec 3.3, we have also modified the notation to make the expression clearer.
>
> Q4: ‘the first work to apply MLP-based architecture to the downstream task" - There seems to be a strong relation to the recent CycleMLP, which may be concurrent work, but should nevertheless be cited and contrasted.’
>
> A4: Thank you for referring to CycleMLP. Similarly, according to the ICLR review rules, we are the concurrent work. More importantly, to the best of our knowledge, our work is the first MLP-based architecture to be applied to the downstream tasks. The main text in our original version is not overclaimed. According to your suggestion, in the revised version, CycleMLP is cited and discussed.

---

> > ### Comment · Reviewer_uLzi · 2021-11-19
> > **Response to response**
> >
> > Thank you for your detailed reply and for making the changes to the manuscript. I updated my main review (see above).

---

> > > ### Author Response · Authors · 2021-11-20
> > > **Response to Reviewer uLzi**
> > >
> > > Thanks for your recognition and raising your score. If you have other concerns, please feel free to reply.

---

> ### Author Response · Authors · 2021-11-16
> **Response to Reviewer uLzi (2/2)**
>
> Q5: ‘Footnote 2: This is a very strong statement ("cannot"). It is probably clear that this does not work "out of the box". This statement could be discussed and explained in more detail.’
>
> A5: Thank you for your suggestion. Now we have modified the statement and moved it to the main text. The modified sentence is as follows: ‘It is also worth noting that the model weights in MLP-Mixer trained with fixed image size cannot be adapted to downstream tasks with various input sizes because the token-mixing MLP has a fixed dimension.’
>
> Q6: ‘4.1 Settings: among the myriad of regularization strategies that exist, why are specifically label smoothing and DropPath chosen? How many other regularization strategies were tried? Or maybe is this based on Swin, in which case it should be noted here.’
>
> A6: Thank you for your suggestion. In the original version, we follow Swin's training strategy. To make it clearer, we rewrite this paragraph in Sec 4.1 (setting) of the revised version.
>
> Q7: ‘4.2 This Section seems more like a parameter choice discussion than an ablation study to me.’
>
> A7: Thank you for your valuable suggestion. We have modified this section title to the choice and impact of AS-MLP block and added comparison experiments to perform the impact of AS-MLP block. See Sec. 4.2 and Table 4 (The impact of AS-MLP block).
>
> Q8: ‘4.5 The statements derived from the four examples in Figure 4 ("From Figure 4, one can see that") seem like overclaiming, at least in parts: "These phenomena further show the superiority of AS-MLP.’
>
> A8: Thank you for your suggestion. We have revised the statement.
>
> Q9: ‘In the context of "axial attention" (even though the AS-MLP does not use attention) it seems there are at least two papers that are quite well-known and it might be useful to add them to the related work: Wang et al.: Axial-DeepLab [...], ECCV, 2020 Ho et al.: Axial Attention in multidimensional transformers, arXiv:1912.12180, 2019’
>
> A9: We have cited these papers in the revised version.
>
> Q10: ‘Minor details’
> A10: Thanks for your careful review. We have improved our paper according to your suggestions. If there are some other problems, you are also welcome to point them out.
>
> If we have misunderstood your questions or you have any other concerns, please feel free to reply. Your constructive comments are very important for the enhancement of our manuscript.

---

### Official Review · Reviewer_HTjG · 2021-11-02

**Correctness:** 3
**Technical Novelty And Significance:** 1
**Empirical Novelty And Significance:** 2
**Recommendation:** 5
**Confidence:** 5

**Main Review:**

Strengths:
The work shows good results on different vision tasks.


Weaknesses:

1. The novelty of the work is quite limited. The idea of shifting operation in vision has been already explored in numerous previous works, such as TSM (Lin et al., 2019) and S2-MLP (Yu et al., 2021). Especially, the proposed work is very similar to S2-MLP and, furthermore, the authors proposed even an improved version, S2-MLPV2, which reports even better results for 224x224 image resolution.

2. The presentation of the results is not fair. For instance, in the ablation study (Table 3a) the authors present the proposed approach of shifting operation as superior comparing it without shifting operation. However, when the shift size is (1,1)  there is no spatial interaction in the architecture (all the information exchange is between the channels at the same spatial location). The authors claimed that introducing locality into MLP-based architecture “enables the model to obtain more local dependencies, thus improve the performance.” But the authors did not actually prove this very clearly. I think in this ablation study, it is necessary to include also the case when using a MLP along spatial dimension. For instance, there can be two cases, first, using  a global MLP along full spatial size, and second, using two MLPs (one MLP along horizontal spatial direction and another MLP along vertical spatial direction). Otherwise, it can be the case that big part of the improvement comes actually from just applying a standard MLP on the horizontal spatial direction and another MLP on the vertical direction.

**Summary Of The Paper:**

The work proposed an approach to introduce locality into the MLP-based architecture by using a shifting operation along vertical and horizontal directions.

**Summary Of The Review:**

The novelty of the work is limited and the authors did not show the results when using a standard MLP approach on the spatial direction.

---

> ### Public Comment · ~TAN_YU2 · 2021-11-15
> **Comments on the review**
>
> Hi, Reviewer HTjG
>
> I am the author of S2-MLP and S2-MLPv2.
> I agree with you that the proposed model has some relation with my work S2-MLP and S2-MLPv2,  but I think the submitted manuscript has one main contribution: it is the earliest work for applying the MLP-based architecture for object detection and segmentation.
> Meanwhile, the spatial shifting operation in the submitted manuscript has some differences from the shiting operations in S2-MLP and S2-MLPv2. Besides,  S2-MLPv2 is still under review, it is not necessary for the submitted manuscript to beat the performance of S2-MLPv2.
> You give only 1 score for **Technical Novelty And Significance** and **Empirical Novelty And Significance**, which I believe is far from fair.

---

> > ### Author Response · Authors · 2021-11-20
> > **Response to the author of S2-MLP**
> >
> > Thanks for your recognition. We have also cited S2-MLPv2 in the revised version. If you have any suggestions, please feel free to reply.

---

> ### Author Response · Authors · 2021-11-16
> **Response to Reviewer HTjG**
>
> Thank you for your insightful and valuable suggestions. We have tried our best to improve our paper by your comments into consideration. Please see the responses as follows.
>
> Q1: ‘The idea of shifting operation in vision has been already explored in TSM (Lin et al., 2019) and S2-MLP (Yu et al., 2021). The proposed work is very similar to S2-MLP and S2-MLPV2, which reports even better results for 224x224 image resolution.’
>
> A1: We first list the main differences between AS-MLP and TSM as follows:
> TSM uses the shift strategy in the temporal dimension and applies it to convolution. We verify the effectiveness of the axial shift module in the spatial dimension and apply it to a pure MLP architecture. The introduction of TSM paper argues that ‘shifting too many channels in a network will significantly hurt the spatial modeling ability and result in performance degradation.’. See Table 3(a) of our original version, as the shift size increases, the channel needs to be divided into more parts, but the performance does not decrease significantly. This also shows that the argument of TSM is not obvious in the pure MLP architecture, which also illustrates the importance of our exploration for pure MLP architecture.
>
> As for S2-MLP and S2-MLPv2, they are indeed excellent work. According to the ICLR review rules (https://iclr.cc/Conferences/2022/ReviewerGuide), they are considered to be concurrent work as pointed in Sec. 2 of our original version. Nevertheless, encouraged by the ICLR community, we cite S2-MLP and discuss the method in our original version. According to your valuable suggestion, we also cite S2-MLPv2 and discuss the method in our revised version (See Sec. 2 for details). More importantly, we are also working on downstream tasks, such as object detection and segmentation, which are not conducted in S2-MLP.
>
> Q2: ‘The presentation of the results is not fair. When the shift size is (1,1) there is no spatial interaction in the architecture.’
>
> A2: We would like to emphasize that our comparison is fair, but there might be unclear statements that mislead you. We introduce shift size (1, 1) in Table 3(a) to show a complete experimental result, and it is also elaborated in Table 3. The caption of Table 3 indicates that case (1, 1) is only the channel dimension interaction. Perhaps this sentence misleads you, but our original intention is to provide experiments of different network configurations. We have modified this description in the revised version. (See Sec. 4.2)
>
> Q3: ‘The authors claimed that introducing locality into MLP-based architecture “enables the model to obtain more local dependencies, thus improve the performance.” The authors did not actually prove this very clearly. There can be two cases, first, using a global MLP along full spatial size, and second, using two MLPs (one MLP along horizontal spatial direction and another MLP along vertical spatial direction).’
>
> A3: For the sentence “enables the model to obtain more local dependencies, thus improve the performance.”, the aim is to compare our work with MLP-Mixer that introduces the global receptive field. We focus on exploring the local receptive field, which is a simple yet crucial component for the performance improvement of MLP based architecture. Our performance in Table 1 has shown the advantage of the local receptive field compared with MLP-Mixer. AS-MLP-S achieves 83.1% Top-1 accuracy but Mixer-B/16 achieves 76.4% Top-1 accuracy with the comparable number of parameters and FLOPs. Regarding the two situations you mentioned, the first situation is similar to the MLP-Mixer. Therefore, following your experiment guidance, we conduct experiments to evaluate the impact of AS-MLP block (See Sec. 4.2 of the revised version). We introduce the global MLP to our architecture. The first situation is Global-MLP in Table 4, and the second situation is Axial-MLP. However, it should be noted that in these two cases, the model parameters cannot be directly transferred to downstream tasks for pretraining because the model weights are limited to a fixed dimension. Furthermore, we also design the following baselines: Window-MLP, shift size (1, 5) and shift size (5, 1), whose settings are detailed in Appendix A.5. As shown in Table 4, our AS-MLP block (5, 5) achieves 81.34% Top-1 accuracy, which outperforms other baselines (Global-MLP: 79.81%, Axial-MLP: 79.69%).
>
> Q4: About 1 score in ‘Technical and Empirical Novelty’.
>
> A4: We have reconfirmed our methodological and empirical contributions in Response to AC and all reviewers, and in the responses above. Furthermore, we have conducted a large number of experiments in the main text and supplementary material, and we consider them to be significant.
>
> If we have misunderstood your questions or you have any other concerns, please feel free to reply. Your constructive comments are very important for the enhancement of our manuscript.

---

> > ### Comment · Reviewer_HTjG · 2021-11-19
> > **response to authors**
> >
> > Thank you for the answers. I still consider that my initial general recommendation is right. The novelty of the work is very limited, the arguments of the authors comparing their work with TSM are not convincing. In my option, the proposed idea is just a special case of TSM. Furthermore, regarding the comment with convolution, the authors also implement the proposed approach using a 1x1 convolution. However, it is a good experimental contribution to extend the MLP-based approaches to object detection and segmentation tasks, but I still question if this is enough for a full paper at the level of ICLR (maybe it is more suitable as a workshop contribution). Thus, I will keep my initial recommendation.

---

> > > ### Author Response · Authors · 2021-11-20
> > > **Response to Reviewer HTjG**
> > >
> > >
> > > Thanks for your new responses. Considering your comments, we elaborate the differences between AS-MLP and TSM in the revised paper (See Appendix A.6). Our responses are as follows.
> > >
> > > Q1: ‘In my option, the proposed idea is just a special case of TSM’
> > >
> > > A1: We argue that our AS-MLP is not a special case of TSM for the following three reasons.
> > > i) TSM performs a shift in the temporal dimension and, as stated in TSM paper, they target the temporal dimension for efficient video understanding. However, we explore the shift from a spatial perspective, for the more general tasks, such as image classification, object detection, and segmentation. TSM is not designed for general tasks, and it cannot be expressed in the principle that AS-MLP is a special case of TSM.
> > > ii) As stated in our first response, the introduction of TSM paper argues that ‘shifting too many channels in a network will significantly hurt the spatial modeling ability and result in performance degradation.’. See Table 3(a) of our original version, as the shift size increases, the channel needs to be divided into more parts, but the performance does not decrease significantly. This suggests that the argument of TSM is not obvious in the pure MLP architecture. In other words, it cannot be assumed that using TSM directly in the spatial dimension will work.
> > > iii) Our motivation is quite different. TSM is designed to be more effective and efficient on video. Our motivation is derived from Swin Transformer’s exploration of the local receptive field of the transformer. We use shift to explore the local receptive field of MLP. Also, AS-MLP is the first MLP-based architecture for object detection and segmentation with the help of such a method. Furthermore, we use axial shift to reduce the complexity of the shift split, as shown in Response to Reviewer uLzi and the introduction of our revised version.
> > >
> > > Q2: ‘Regarding the comment with convolution, the authors also implement the proposed approach using a 1x1 convolution.’
> > >
> > > A2: We perform 1x1 convolutions to implement the MLPs. However, there is an inherent difference between 1x1 convolution and 3x3 convolution. A 1x1 convolution is an MLP and does not have a spatial receptive field (or 1x1 receptive field), but a 3x3 convolution, which has a 3x3 receptive field and is regarded as the core of the previous convolution network family, such as ResNet or DenseNet. By stacking 3x3 convolutions, the model can obtain a larger receptive field. This also means that with only 1x1 convolutions, the performance of the model will be extremely degraded, as has been shown in some previous conv-based architectures. TSM has also designed Residual TSM module with 3x3 convolution to improve performance (e.g. Figure 3 in the TSM paper). However, using 1x1 convolutions (MLPs), we are able to obtain comparable results with the transformer-based architecture, which is impressive and shows the contribution of our work. We also design a simple and efficient way to implement axial shift in a few lines of code, and we provide a cuda implementation that makes the complexity of the shift operation independent of the shift size.
> > >
> > > We have also listed our contributions and strengths in Response to AC and all reviewers. We have also restated your points of concern and revised the manuscript again. We hope that this response has addressed your concerns. We also believe that our revised manuscript is well-qualified for this conference.
> > >
> > > Another, do our previous responses and revised manuscript address your other concerns? Would you like to raise your score?

---

### Author Response · Authors · 2021-11-16
**Response to AC and all Reviewers**

Dear AC and all Reviewers,
We are grateful to the three reviewers for their valuable comments and recognition of our work. Our previous version might have some ambiguities in the description of some details. Currently, we have updated the version with more explanations to make the discussion clearer. Reviewers might have the following concerns, such as i) the motivation of the method; ii) the novelty of the method, and iii) the design of some baseline experiments for comparisons. We have thoroughly revised our manuscript according to the comments made by the reviewers to make our work well-qualified.

We would like to highlight the advantages of AS-MLP as follows:
i) inspired by the design of the window receptive field in Swin Transformer (ICCV21 best paper), we explore the importance of the local receptive field in MLP-like architectures compared to the global receptive field in MLP-Mixer.
ii) we propose the axial shift operation to implement local receptive fields in a simple and efficient way, and it is also the first MLP-like architecture that can adapt to different input dimensions for object detection and segmentation.
iii) AS-MLP outperforms all MLP-based architectures and achieves competitive performance compared to the transformer-based architectures even with slightly fewer parameters.

Note: S2-MLP, S2-MLPv2, Vip and CycleMLP are considered to be the concurrent work, according to the ICLR review rules (https://iclr.cc/Conferences/2022/ReviewerGuide). Encouraged by the ICLR community, we have cited all the above papers and introduced their methods. We also conduct experiments for comparisons with some of these papers, which would further strengthen our work and facilitate readers' better understanding of the progress of the area. Our revised manuscript will be well-qualified for this conference.

We have posted responses under each reviewer’s window and revised the manuscript. All changes are highlighted in magenta and we hope that the revised version addresses your concerns. If AC and reviewers have other questions, we are willing to friendly discuss them.

---

### Decision · Program_Chairs · 2022-01-20

**Decision:**

Accept (Poster)

**Comment:**

The paper proposes a MLP-based architecture that makes extensive use of the shift operation on the feature maps. The model performs well on several vision tasks and datasets.

The reviews are mixed even after the authors' response. Main pros are that the proposed architecture is elegant and reasonable, and the experimental evaluation is thorough and strong. The main con is that the novelty is somewhat limited to some prior papers.

Overall, I recommend acceptance. The reviewers point out that the architecture is good and the results are strong. Similarities to prior works do not seem serious enough to warrant rejection - even an author of arguably the most related (concurrent) works - S2-MLP and S2-MLPv2 - confirms that there is sufficient difference. Moreover, this is one of the first papers to show very strong results on detection and segmentation.